# Host Immune Response Driving SARS-CoV-2 Evolution

**DOI:** 10.3390/v12101095

**Published:** 2020-09-27

**Authors:** Rui Wang, Yuta Hozumi, Yong-Hui Zheng, Changchuan Yin, Guo-Wei Wei

**Affiliations:** 1Department of Mathematics, Michigan State University, East Lansing, MI 48824, USA; wangru25@msu.edu (R.W.); hozumiyu@msu.edu (Y.H.); 2Department of Microbiology and Molecular Genetics, Michigan State University, East Lansing, MI 48824, USA; zhengyo@msu.edu; 3Department of Mathematics, Statistics, and Computer Science, University of Illinois at Chicago, Chicago, IL 60607, USA; cyin1@uic.edu; 4Department of Biochemistry and Molecular Biology, Michigan State University, East Lansing, MI 48824, USA; 5Department of Electrical and Computer Engineering, Michigan State University, East Lansing, MI 48824, USA

**Keywords:** SARS-CoV-2, COVID-19, APOBEC, ADAR, gene editing

## Abstract

The transmission and evolution of severe acute respiratory syndrome coronavirus 2 (SARS-CoV-2) are of paramount importance in controlling and combating the coronavirus disease 2019 (COVID-19) pandemic. Currently, over 15,000 SARS-CoV-2 single mutations have been recorded, which have a great impact on the development of diagnostics, vaccines, antibody therapies, and drugs. However, little is known about SARS-CoV-2’s evolutionary characteristics and general trend. In this work, we present a comprehensive genotyping analysis of existing SARS-CoV-2 mutations. We reveal that host immune response via APOBEC and ADAR gene editing gives rise to near 65% of recorded mutations. Additionally, we show that children under age five and the elderly may be at high risk from COVID-19 because of their overreaction to the viral infection. Moreover, we uncover that populations of Oceania and Africa react significantly more intensively to SARS-CoV-2 infection than those of Europe and Asia, which may explain why African Americans were shown to be at increased risk of dying from COVID-19, in addition to their high risk of COVID-19 infection caused by systemic health and social inequities. Finally, our study indicates that for two viral genome sequences of the same origin, their evolution order may be determined from the ratio of mutation type, C > T over T > C.

## 1. Introduction

The ongoing outbreak of coronavirus disease 2019 (COVID-19), caused by severe acute respiratory syndrome coronavirus 2 (SARS-CoV-2), has led to tremendous human mortality and economic hardship. As of 31 July 2020, over 17,106,007 confirmed COVID-19 cases have been reported worldwide and 668,910 deaths have occurred from the disease [1]. To mitigate this devastating pandemic, we have to control its spread by sufficient testing, social distancing, contact tracking, and developing effective diagnosis tools, efficacious antiviral drugs, antibody therapies, and preventive vaccines.

SARS-CoV-2 is a positive-sense, single-strand RNA virus that belongs to the beta coronavirus genus [2]. It has a genome size of 29.82 kb, which encodes multiple non-structural and structural proteins. The ORF1ab encode non-structural proteins for RNA replication and transcription. The downstream regions of the genome encode structural proteins, including the spike (S) protein, the nucleocapsid (N) protein, the envelope (E) protein, and the membrane (M) protein. All of the four major structural proteins are required to produce a structurally complete viral particle. The S protein mediates viral attachment to the host angiotensin-converting enzyme 2 (ACE2) receptor and subsequent fusion between the viral and host cell membranes aided by transmembrane serine protease 2 (TMPRSS2) to allow the entry of viruses into the host cell [3,4,5]. The nucleocapsid (N) protein, one of the most abundant viral proteins, binds to the RNA genome and is involved in replication processes, assembly, and host cellular response during viral infection [6].

Mutagenesis is a basic biological process that changes the genetic information of organisms. As a primary source of many kinds of cancers and heritable diseases, mutagenesis may be fearful but is a driving force for natural evolution [7,8]. Although viruses are not organisms per se, they are at the edge of life. Our real-time interactive SARS-CoV-2 Mutation Tracker (https://users.math.msu.edu/users/weig/SARS-CoV-2_Mutation_Tracker.html) shows that over 15,000 mutations have occurred on SARS-CoV-2 [9]. More than 1700 mutations on the S protein gene have a significant impact on SARS-CoV-2 infectivity [10,11,12]. These mutations should be put into the perspective that COVID-19 has globally spread. The geographical and demographical diversity of the viral transmission and exogenous and endogenous genotoxins exposure have stimulated SARS-CoV-2 mutations. If we consider the average number of mutations per genome, SARS-CoV-2 is mutating slower than other viruses, such as the flu and common cold viruses [13,14]. This is because SARS-CoV-2 belongs to the coronaviridae family and the Nidovirales order, which has a genetic proofreading mechanism in its replication achieved by an enzyme called non-structure protein 14 (NSP14) in synergy with NSP12, i.e., RNA-dependent RNA polymerase (RdRp) [15,16]. As a result, SARS-CoV-2 has a relatively high fidelity in its transcription and replication processes. In general, Coronavirus mutations are created from three major sources, namely, random errors in replication, such as genetic drift and spontaneous genotoxins, viral replication proofreading and defective repair mechanisms, and host immune responses, such as gene editing [11,17]. Genotyping tracks mutations, overpopulation, space, and time, while also providing a method to understand the molecular mechanism of SARS-CoV-2 proteins, protein–protein interactions, and their synergy with host cell proteins, enzymes, and signaling pathways.

The studies of SARS-CoV genomes have, to date, predominantly focused on understanding genome mutation variants, implications in virus transmissions [18,19], and ramifications on the development of diagnostics [9,20], vaccines [21], antibodies [22], and drugs [21]. Although it is difficult to determine the detailed mechanism of every specific mutation, early work on a few initial SARS-CoV-2 strains in Wuhan, China, revealed that hypermutations C > T are most likely resulted from the APOBEC (apolipoprotein B mRNA editing enzyme, catalytic polypeptide-like) deamination in RNA editing [23]. Simmonds noted that a large proportion of C > T mutations in a host APOBEC-like context in the initial months of the pandemic provide evidence for a potent host-driven antiviral editing mechanism against SARS-CoV-2 [24]. An accumulation of C > T mutations in SARS-CoV-2 variants was also reported in [25]. In the standard genetic code, all three stop codons, TAA, TAG, and TGA, involve T but not C. Therefore, the gene-editing imposed C > T mutations will have a high possibility of terminating the translation of viral proteins, which undermines viral functions and survivability. To be noted, as stated in [25], the C > T mutation can increase the frequency of codons for hydrophobic amino acids, which may have an effect on the properties of SARS-CoV-2 proteins. The spontaneous C > T transitions and APOBEC deamination can lead to cancer in humans. There are two well-known deaminase RNA editing mechanisms in human cells: the APOBEC [26] and the adenosine deaminases acting on RNA (ADAR) [27]. The APOBEC enzymes deaminate cytosines into uracils (C > U) on single-stranded nucleic acids (ssDNA or ssRNA). It is well established that the human genome encodes activation-induced cytidine deaminases (AIDs) and several homologous APOBEC cytidine deaminases that function in innate immunity as well as in RNA editing [28,29]. In both innate and adaptive immunity, AID and APOBEC cytidine deaminases modulate immune responses by mutating specific nucleic acid sequences of hosts and pathogens. The ADAR enzymes deaminate adenines into inosines (A-to-I) and result in A > G mutation. The significance of A-to-I editing is its abundance in both host and viral RNAs. ADAR enzymes play important roles during viral infections. They can have either a proviral or an antiviral consequence, depending on the virus–host combination [30,31].

The APOBEC family proteins play critical functional roles within the adaptive and innate immune system, which is involved soon after the infection [32]. Therefore, the higher ratio of C > T mutations may indicate the strong capacity of the host immune system. However, a strong immune response is a double-edged sword. On the one hand, it may help host cells to defeat the virus more efficiently. On the other hand, it can result in a “cytokine storm”, which is a key cause of the death of COVID-19 patients by the exponential growth of inflammation and organ damage [33].

In this work, we analyze a large volume of single-nucleotide polymorphisms (SNPs) found in 33,693 complete SARS-CoV-2 genome isolates globally. By analyzing the distribution of 12 SNP types, we notice that the ratio of C > T mutations is predominately higher than that of the other types of mutations, indicating that hypermutation C > T may result from extensive host RNA editing, i.e., the APOBEC deamination. Additionally, we investigate the distribution of 12 SNP types in different age groups, gender groups, and geographic locations to understand whether these hypermutations have a age/gender/demographic preference. Moreover, we provide deep insights into the mutation motif and hot-spot patterns from 13,833 single mutations decoded from 33,693 complete SARS-CoV-2 genome sequences, revealing mutational signatures and preferred genetic environments. Finally, we hypothesize that virus genomes evolve through host innate immune response imposed gene editing, i.e., C > T, and virus protective mechanism-installed defective revisionary mutations, T > C. As a result, both C > T and T > C mutation ratios are usually high. We show that the ratio of C > T to T > C mutations is higher than the unity in the forward viral evolution, which suggests the master and slave relationship between host gene editing and virus protective mechanism. Therefore, we propose that the C > T to T > C ratio being higher than the unity (>1) is an indication of the forward viral evolution direction.

## 2. Methods and Materials

### 2.1. SNP Genotyping

Here, 33,693 complete genomes of the SARS-CoV-2 strains of the globe were retrieved from the GISAID database [34] on 31 July 2020. Only the complete genomes of high-coverage that have no stretches of ’NNNNN’ were included in the dataset. The complete genome sequences were aligned with the reference genome of SARS-CoV-2 by the multiple sequence alignment (MSA) tool Clustal Omega using the default parameters [35]. The SNP mutations were retrieved from the aligned genomes according to the reference SARS-CoV-2 genome that is derived from the human coronavirus Wuhan-1 isolate (GenBank access number: NC_045512.2) [2]. The SNP profile, including nucleotide changes and the corresponding positions in a genome, can be considered as the genotype of the virus.

### 2.2. Multiple Sequence Alignment

As mentioned above, Clustal Omega is used to do multiple sequence alignments. The basic idea of the Clustal Omega is to produce a pairwise alignment using the *k*-tuple method. Given two sequences, $S_{1}$ and $S_{2}$, a *k*-tuple match is defined as *j*-letters that match in $S_{1}$ and $S_{2}$
(1)$$\left\{S_{1}\left(i\right),S_{1}\left(i+1\right),...,S_{1}\left(i+k-1\right)\right\}=\left\{S_{2}\left(j\right),S_{2}\left(j+1\right),...,S_{2}\left(j+k-1\right)\right\}$$
where i,j are positions of the sequence, such as DNA, RNA and amino acid sequences. The distance between the two sequences is then defined as
(2)$$d_{i}\left(S_{1},S_{2}\right)=1-\frac{C_{i}\left(S_{1},S_{2}\right)}{C_{t}},$$
where $C_{i}\left(S_{1},S_{2}\right)$ is the number of *k*-tuple matches between $S_{1}$ and $S_{2}$ and $C_{t}$ is the sum of all paired residue of $S_{1}$ and $S_{2}$ (i.e., the total number of matches). A dot matrix between $S_{1}$ and $S_{2}$ can be computed by matching *k*-tuples between $S_{1}$ and $S_{2}$. The diagonal element above a certain threshold is called the “window space”. By utilizing the Needleman–Wunsch method restricted to the “window space”, we can find an alignment, which has the highest score that can be calculated from the Needleman–Wunsch method [35].

To find the optimal alignment, other methods, such as the mBed method and the UPGMA method, can also be employed [36,37]. As an advanced MSA tool, the Clustal Omega guarantees high accuracy and is time-efficient.

### 2.3. SNP Analysis

The Cluster Omega is employed to carry out the multiple sequence alignment. The genomic analytics are performed using computer programs in Python and Biopython libraries [38].

### 2.4. Data Availability

The nucleotide sequences of the SARS-CoV-2 genomes used in this analysis are available, upon free registration, from the GISAID database (https://www.gisaid.org/). The SNP IDs and the acknowledgments of the SARS-COV-2 genomes are given in the Appendix A.

## 3. Results

To reveal that C > T and A > G mutations are driven by RNA-APOBEC and RNA-ADAR editing, we first analyzed 33,693 complete SARS-CoV-2 genome sequences, and a total of 13,833 single mutations were found as of 31 July 2020. To be noted, 13,833 single mutations were unique mutations, i.e., the same mutation appearing in different SARS-CoV-2 isolates is only counted once. If we count the same mutation in different SARS-CoV-2 isolates repeatedly according to their frequency, then all of the mutations that are detected in the 33,693 complete SARS-CoV-2 genome sequences are called non-unique mutations. With the reference sequence of SARS-CoV-2 genome collected on 5 January 2020 [2], we calculated the proportion of 12 SNP types (i.e., A > T, A > C, A > G, T > A, T > C, T > G, C > T, C > A, C > G, G > T, G > C, G > A) worldwide. The unusually high ratios of C > T and A > G mutations indicate that RNA-APOBEC editing and RNA-ADAR editing are involved in the host immune response to SARS-CoV-2 infection [28,29]. Additionally, to understand gene-editing preference, we investigated the distribution of 12 SNP types of mutations in different countries/regions, age groups, and gender groups. Furthermore, we decoded mutation motifs from the 2-mer and 3-mer sequence contexts to survey the hot-spot patterns and mutational signatures driven by gene-editing. Moreover, we analyzed the proportion of 12 SNP types among SARS-CoV, Bat-SL-BM48-31, Bat-SL-CoVZC45, Bat-SL-RaTG13, and SARS-CoV-2. We discovered that the viral evolution order can be determined by the ratios of C > T/T > C. These results are presented in following subsections.

### 3.1. Host Immune Response to SARS-CoV-2 Infection with Gene Editing

#### 3.1.1. Global Analysis

Table 1 illustrates the proportion of 12 SNP types of SARS-CoV-2 (i.e., A > T, A > C, A > G, T > A, T > C, T > G, C > T, C > A, C > G, G > T, G > C, G > A) in the global. Here we only consider the unique SNPs.

First, it can be seen that not all SARS-CoV-2 mutations are created equal. Mutation C > G only accounts for 1.25%. A few other mutation types, G > C, T > G, T > A, and A > C, are not frequent either. If mutations are random, each mutation should have a ratio of 8.3% on average. It can be seen that C > T owns the largest proportion (24.06%), which is much higher than the average ratio. Applying the Z-test, the *p*-value is $2.74\times 10^{-274}$, which means our deduction is of statistical significance. Therefore, the hypermutation C > T must be driven by additional mechanisms. It is known that host RNA-APOBEC editing leads to excessive C > T transitions.

Moreover, the second most frequent mutation type is A > G transition. Its A > G ratio of 14.87% is much higher than the average ratio of 8.3%, and the *p*-value $7.9\times 10^{-64}$ from the Z-test implies that our deduction passes the hypothesis test. Therefore, RNA-ADAR editing is also involved in the host immune response. Although the high ratios of C > T and A > G reveal that the immune system is combating SARS-CoV-2 by two deaminase RNA editing mechanisms, the relatively high ratios of the reversed mutations T > C and G > A also indicate that SARS-CoV-2 fights the gene editing using its defective proofreading and repairing mechanisms.

Finally, it is well-known that mutations can be classified into four transition types (i.e., A > G, G > A, C > T, and T > C) and eight transversion types. Table 1 shows that all transition types have relatively high ratios, whereas all transversion types, except for G > T, have relatively low ratios. This is due to the fact that it is easier to substitute a single-ring nucleotide structure for another single-ring nucleotide structure than to substitute a double-ring nucleotide for a single-ring nucleotide. Additionally, transitions are more likely to result in silent mutations. Therefore, transversions can be more destructive to viral genomes.

#### 3.1.2. Age Analysis

Figure 1 illustrates the distribution of 12 SNP types among unique SNPs in SASR-CoV-2 genome isolates from different age groups. In general, with the increase in age, the ratio of C > T gradually increased. Here, 42.1% C > T mutations were detected in patients who are older than 90 years old, indicating that gene-editing is more active in elderly patients than young patients. We applied the Z-test to the data of patients older than 90 years old and patients in other age groups, and all the resulting *p*-values were less than 0.05. Therefore, our hypothesis is statistically significant. However, the severe COVID-19 cases may be due to the immune systems’ heightened response. When SARS-CoV-2 infects a host cell, a set of proteins called cytokines will be released from a broad range of cells (mainly immune cells). Cytokines are involved in the immune response to produce more immune cells and recruit them to the sites of inflammation in order to fight against the viral infection. In turn, more cytokines can be released from the immune cells. This positive feedback loop will result in a “cytokine storm”, which can beget the exponential growth of inflammation, trigger apoptosis, and lead to organ damage [33]. Therefore, we hypothesize that if the immune system overreacts to the invading pathogens, it is more likely to cause the cytokine storm and aggravate the condition of COVID-19 patients. It can be seen in Figure 1, where patients who are older than 80 years old have more C > T mutations compared to other age groups. This result reveals that APOBEC3 gene editing is more active with older people. Consequently, the cytokine storm may happen more frequently in older people than it does in younger people. This might be one of the main causes of the high COVID-19 fatality for the elderly. Age-related mutagenesis, i.e., C > T transition, is known to cause more cancer diagnostics in the elderly [39].

Notably, the SARS-CoV-2 samples from children under five years old have a relatively high ratio of C > T mutations (39.6%). The *p*-value from the Z-test is less than 0.05, which also implies the statistical significance of a relatively high ratio of C > T in patients under five years old. Moreover, the reversed mutation type T > C for samples from children under five years old and adults older than 90 years old has the third-largest ratio. In other age groups, T > C has the fourth-largest ratio. As demonstrated before, the reversed mutation T > C may reveal that SARS-CoV-2 is capable of fighting back against the host immune system. Therefore, we deduce that SARS-CoV-2 will fiercely counter-attack the immune system in children under five and adults older than 90.

Our result reveals that the immune systems of children under five years old are less well-developed and weaker than those of adults. They have to fight harder when SARS-CoV-2 infects. This result suggests that children under five are at risk of COVID-19. However, the long-term health consequence of young children’s unusual response to SARS-CoV-2 infection remains to be further studied.

#### 3.1.3. Gender Analysis

Figure 2 shows the distribution of 12 SNP types in SARS-CoV-2 genome isolates globally from two gender groups. The ratio of C > T mutations in females is slightly higher in males. We also apply a simple Z-test, and the *p*-value is 0.0302, which is smaller than the significance level 0.05. Therefore, the slightly higher C > T ratios in females matches the finding that women have a stronger immune response than men [40,41]. Moreover, Figure 3 and Figure 4 depict the distribution of 12 SNP types in different age groups among female and male patients. Overall, the proportion of C > T mutations in the SARS-CoV-2 genomes from females is higher than the C > T proportion in the SARS-CoV-2 genomes from males, except for ages between 6 and 19 and ages older than 90. Therefore, we can deduce that the RNA editing has an age and gender preference, and is therefore more likely to happen or become stronger for females who are older than 90 years old or under 5 years old.

#### 3.1.4. Geographic Analysis

In this section, we analyze the distribution of SARS-CoV-2 mutations in different countries and regions. Limited by the number of complete genome sequences submitted to GISAID that have appropriate labels, we only analyzed the countries with more than 1000 labeled sequences to maintain statistical significance. Table 2 lists the total number of SARS-CoV-2 sequences in the United Kingdom, United States, Australia, and India. The number of sequences with age and gender information is given in Table 2.

Table 3 illustrates the C > T ratios in the SARS-CoV-2 genome isolates from different age groups in the United Kingdom, United States, Australia, and India, respectively. We can see that the SARS-CoV-2 genome isolates from the United Kingdom patients have the highest ratio of C > T compared to those from the other three countries. It is interesting to note that the SARS-CoV-2 genome isolates from the patients older than 80 years old from the United Kingdom and Australia have less C > T mutations, which is not consistent with the global pattern. The distribution of 12 SNP types in the SARS-CoV-2 genome isolates from different age groups in the United Kingdom, United States, Australia, and India can be found in the Appendix A.

Figure 5 illustrates the distribution of 12 SNP types in six continents. The SARS-CoV-2 genome isolates from Europe, Asia, and North America patients have a relatively low C > T mutation ratio (less than 35%), while the reversed T > C mutation ratio is relatively high (greater than 10%). When employing the Z-test between the data in the world and in Europe, Asia, and North America, the *p*-values are all less than $10^{-5}$, which means that the C > T and T > C ratios in Europe, Asia, and North America are not at the same level as the global ratio. On the contrary, South America, Oceania, and Africa have higher C > T ratios but a lower T > C ratio. It worth noting that the C > T mutation ratios in the SARS-CoV-2 genome isolates from Oceania and Africa are more than 10% higher than those of Asia, Europe, and North America. This result indicates that the APOBEC editing may be more active, and the counterattack of SARS-CoV-2 might be weakened by the strong immune response in the populations of Oceania and Africa.

African Americans, as an ethnic group of Americans with total or partial ancestry from Africa, are genetically associated with Africans. There have been many concerns about the fact that they are disproportionately affected by COVID-19 (https://www.cdc.gov/coronavirus/2019-ncov/community/health-equity/race-ethnicity.html). The present finding indicates that the immune systems of African Americans may also overreact to SARS-CoV-2 infection by excessive gene editing.

Another interesting issue is that A > G mutation ratios of genome isolates from Oceania and Africa are very low (<9%). In contrast, the A > G mutation ratios of genome isolates from other regions are significantly higher (>11%). The Z-test was also employed to prove that our deduction is statistically significant. For example, when we extracted the number of A > G mutations in Oceania and Asia for calculation, the *p*-value was 0.015, which is less than the significance level 0.05. Similarly, we repeated the procedure. For patients in Oceania and North America, Oceania and South America, Oceania and Europe, all the *p*-values are less than the significance level 0.05. These results indicate that Asia and Europe populations may have adopted significantly different genetic and molecular mechanisms in their immune response to viral infection compared to those of Oceania and Africa. Further studies are required to fully understand these differences.

### 3.2. The SNP Preferences on Sequence Contexts

The mutation preferences in sequence contexts may be used to predict the mutational signatures from genome sequences. Despite numerous studies of the mutation contexts in APOBEC editing inhuman cells, little is known of the mutation contexts in the SARS-CoV-2 genome. As we have a large number of SNP mutations from SARS-CoV-2 genomes, here, we discuss the mutation frequencies from 2-mer and 3-mer sequence contexts. We present 4-mer sequence contexts in the Appendix A.

In general, the patterns discussed in this section are consistent with those presented in Section 3.1.1. However, this section offers more detailed information about mutational signatures.

For mutation motifs of SNPs at the first position of 2-mers Figure 6a, we observe that motif 2-mer CW (where W is either A or T) for the C > T mutation is the predominant context. Similarly, for mutation motifs of the SNPs at the second position of 2-mers Figure 6b, motif 2-mer WC for C > T mutation is the predominant context. These results are consistent with the previous study that TCW contexts (where W = A or T) are predominantly caused by the APOBEC-catalyzed deamination of cytosine (C) to thymine (T) or uracil (U) in human cancer cells [42].

For the SNPs at the first position of 3-mers (ANN or TNN) (Figure 7a), we observe the following mutation patterns.

(1)ANN has high A > G mutations;(2)TNN has a high frequency in T > C mutations.

For the SNPs at the first position of 3-mers (CNN or GNN) shown in Figure 7b, we observe the following mutation patterns.

(1)CNN has a high frequency in C > T mutations;(2)GGA and GGT has a high frequency in G > A mutations;(3)GAA has a relatively high frequency in G > A mutations;(4)GGW (where W is A or T) has relatively high frequency in G > T mutations.

For the SNPs at the second position of the 3-mers (NAN or NTN), as shown in Figure 8a, we observe the following mutation patterns.

(1)NAN has a high frequency in A > G mutations;(2)NTN has a high frequency in T > C mutations;(3)The A > C mutation also has a larger proportion in AAW (where W is A or T).

For the SNPs at the second position of the 3-mers (NCN or NGN), as shown in Figure 8b, we observe the following mutation patterns.

(1)WGN (where W is A or T) has a G > T dominated mutation except for AGG;(2)SGN (where S is G or C) has G > A dominated mutations;(3)AGG has high G > A mutations;(4)Characteristic combinations SCG (where S is G or C) are stable and only a few C > G mutations are detected;(5)Characteristic combinations GGS (where S is G or C) are stable, only a few G > C mutations are detected.

For the SNPs at the third position of 3-mers (NNA or NNT) a shown in Figure 9a, we observe the following mutation patterns.

(1)A > G mutation has a high frequency in NNA;(2)T > C mutation has a high frequency in NNT;(3)T > C mutation is dominated in NGT and only a few T > A and T > G are found in the sequence context of NGT.

For the SNPs at the third position of 3-mers (NNC or NNG) as shown in Figure 9b, we observe the following mutation patterns.

(1)NNC has a high frequency in C > T mutations;(2)G > T mutation has a high frequency in NNG;(3)G > A is also highly expressed in the sequence context of NNG;(4)Characteristic combinations CGC are stable and the mutations on these patterns are most likely to be C > T transitions.

### 3.3. Coronavirus Evolution

It is reasonable to assume that the five coronaviruses, SARS-CoV (2003) [43], Bat-SL-BM48-31 (2008) [44], Bat-SL-CoVZC45 (2017) [45], Bat-SL-RaTG13 (2013) [46] and SARS-CoV-2 (2019) [2], are of the same origin but differ from each other by their evolutionary stages. The data collection date of Bat-SL-RaTG13 (2013) was denoted as 24 July 2013, while the data were not uploaded to the GISIAD database until 27 January 2020. Figure 10 shows the mutation ratio among these five genomes. First, similar to the SARS-CoV-2 mutations listed in Table 1, four transition types (i.e., A > C, C > A, C > T, and T > C) still have high mutation ratios. Notably, the C > T type has the highest ratio, indicating that host immune response still plays the major role. However, transversion type G > T is not as important as in the SARS-CoV-2 mutations discussed earlier. Nonetheless, transversion types A > T and T > A appear on the top six mutation types.

We hypothesize that gene editing via APOBEC (C > T) and ADAR (A > G) is a driving force for RNA viral evolution, as shown in Table 1. Viruses may fight the host immune response with either defective repair or reversed mutations (T > C) within survived isolates. Therefore, the T > C mutation rate would decrease during evolution. We are interested in not only the C > T transition ratio, but also the ratio of C > T over T > C, the reversed transitions. From Figure 10, we can calculate the C > T over T > C ratios among different SARS-like species which are listed in Table 4.

It is seen that viral evolution order may be determined by the T > C over T > C ratio. Through this analysis, we have the following evolution order for the aforementioned coronaviruses: SARS-CoV (2003) → Bat-SL-BM48-31 (2008) → Bat-SL-CoVZC45 (2017) → Bat-SL-RaTG13 (2013) → SARS-CoV-2 (2019) → 33693 SARS-CoV-2 genome isolates (2020). Here, we have one reversed order between Bat-SL-CoVZC45 (2017) → Bat-SL-RaTG13 (2013). This may happen for a few reasons. First, these coronaviruses may not be of the same origin. Second, the data collection date may not be accurate. The sequence of Bat-SL-RaTG13 (2013) was not uploaded until 2020. Finally, our method may admit a few counterexamples.

## 4. Discussion

### 4.1. Comparison of Unique Mutations and Non-Unique SARS-CoV-2 Mutations

The distribution of unique SNP types of 33,693 SARS-CoV-2 isolates is listed in Table 1. The C > T SNP type is remarkably higher than other mutation types. From the distribution of the 12 SNP types, we may infer that the excessive C > T transitions cannot explained by random mutations; instead, hypermutation C > T is due to the cytosine-to-uridine deamination gene editing in human host response. It is interesting to know the distribution of non-unique SNP types.

Figure 11 presents a comparison of the ratios of 12 SNP types among unique and non-unique mutations over all the SARS-CoV-2 genome isolates. The most striking feature is that the C > T ratio is more than doubled in the non-unique mutations, which indicates the overwhelming host immune response to viral infection. Another interesting feature is that the inverse transition T > C has a dramatic reduction of 68% from the unique mutation ratio to the non-unique mutation ratio. These changes reflect the fact that many C > T mutations are high-frequency ones, whereas virus-reversing T > C mutations are low-frequency in nature. The same explanation applies to many mutation types in Figure 11 that have significantly reduced their ratios in the non-unique mutations. However, we observed that the ratios of mutation types A > G, G > T, and G > A do not change much in the non-unique mutations, reflecting the fact that these mutation types maintain a near-average frequency.

Figure 11 shows that the second most frequent mutation type is A > G transitions, standing at 13.6%. The combined C > T and A > G transition types account for nearly 65% of all mutations. Therefore, host gene editing via APOBEC and ADAR is the major driving force of SARS-CoV-2 evolution.

Neutralizing antibodies play a significant role in the clearance of viruses and have been considered a crucial immune artifact for the defense or treatment of viral diseases. However, a clinical study shows that five percent of people recovered from COVID-19 had no detectable antibodies [47]. Another observation is that there are a large number of asymptomatic carrier transmissions of COVID-19 [48]. The reason for the no-antibody COVID-19 recovery and asymptomatic carriers is unknown. From the mutation analysis in this study, APOBEC3 RNA editing is implicated as a strong secondary defense system for the mutating virus and, consequently, for mitigating infection. We postulate that COVID-19 recoveries or convalescents without antibody, and some asymptomatic carriers, are probably due to the increased APOBEC3 activity in host immune systems.

### 4.2. Comparison of Unique Mutations and Non-Unique MERS-CoV-2 Mutations

In this section, we would like to know whether there are similar gene editing behaviors associated with other virus transmission and infection rates. We note that the distribution of SNP types depends on the viral structure, function and its interaction with the host, the degree of infection spread, and the duration of the viral outbreak. Therefore, it is not very meaningful to directly compare the distributions of the SNP types of two viral outbreaks. Nonetheless, we considered the recent Middle East respiratory syndrome–related coronavirus (MERS-CoV) outbreak. We downloaded 523 MERS-CoV sequences from the GenBank. By using the MERS-CoV isolate HCoV-EMC/2012 as the reference (Genbank access number: NC_019843.3), we present the comparison of the ratios of 12 SNP types among unique and non-unique mutaitons over all the MERS-CoV genome isolates in Figure 12. From Figure 11 and Figure 12, it can be seen that the C > T ratio in SARS-CoV-2 isolates is higher than the C > T ratio in MERS-CoV isolates. It is noteworthy that although the C > T ratio is not the highest in the unique case, the C > T is the top-ranked mutation in the non-unique case, indicating that gene-editing may also be active in MERS-CoV infected hosts.

### 4.3. Gene- and Protein-Specific Analysis

Figure 13 illustrates the distribution of mutations on the complete SARS-CoV-2 genome. The *y*-axis represents the natural log frequency for each mutation on a specific position of the complete SARS-CoV-2 genome. We also create a website to report single mutations and update it regularly. We recommend that interested readers go to our website (https://users.math.msu.edu/users/weig/SARS-CoV-2_Mutation_Tracker.html) for detailed information.

Furthermore, we calculated the C > T ratios on all SARS-CoV-2 proteins. Table 5 shows the C > T ratios in 26 SARS-CoV-2 proteins. We can see that NSP9 has the highest C > T ratio, and ORF7b protein has the lowest C > T ratio.

For each gene, the corrected C > T ratio is computed by the original C > T ratio divided by the ratio of C in the gene. We further found that the Pearson correlation coefficient between the C > T ratios and corrected C > T ratios over all genes in the SARS-CoV-2 genome is 0.86, which indicates that the C > T ratio may be not affected much by the ratio of C nucleotides. It is the structure and function of each gene that determine its C > T ratio.

## 5. Conclusions

We use genotyping to analyze the mutation types and their distributions of SARS-CoV-2 genome isolates. We show that host gene editing, namely APOBEC (apolipoprotein B mRNA editing enzyme, catalytic polypeptide-like) and ADAR (adenosine deaminases acting on RNA), are the main driven forces of SARS-CoV-2 evolution, accounting for near 65% of recorded mutations. We reveal that the immune systems of children under age five and the elderly appear to overreact to SARS-CoV-2 infection and may be at high risk from COVID-19. Some minor gender dependence in immune response was also detected. We uncover that the populations of Oceania and Africa react significantly more intensely to SARS-CoV-2 infection than those of Europe and Asia. Our study indicates that while systemic health and social inequities have put African Americans at increased risk of getting sick from COVID-19, their immune systems’ overreaction to viral infection may put them at increased risk of dying from COVID-19. The mutational signatures have been analyzed to explore the preferred gene editing environments. Finally, we show that the ratio of mutation type C > T over T > C may be used to indicate the evolution direction and distinguish the evolution order between two genome sequences of the same origin.

## Figures and Tables

**Figure 1 viruses-12-01095-f001:**
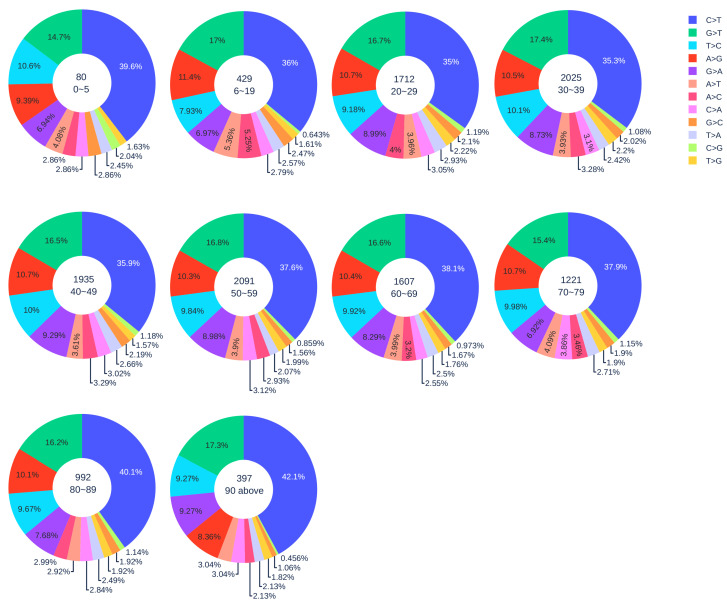
The distribution of 12 SNP types among unique mutations in the SARS-CoV-2 genome isolates from different age groups. The text inside each circle represents the total number of records that have age information for different age groups.

**Figure 2 viruses-12-01095-f002:**
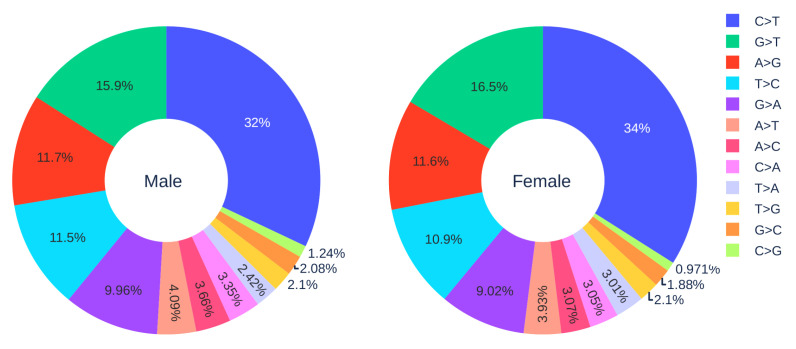
The distribution of 12 SNP types among unique mutations in the SARS-CoV-2 genome isolates from two gender groups. The total number of records in the female group is 5762 and the total number of records in the male group is 7012.

**Figure 3 viruses-12-01095-f003:**
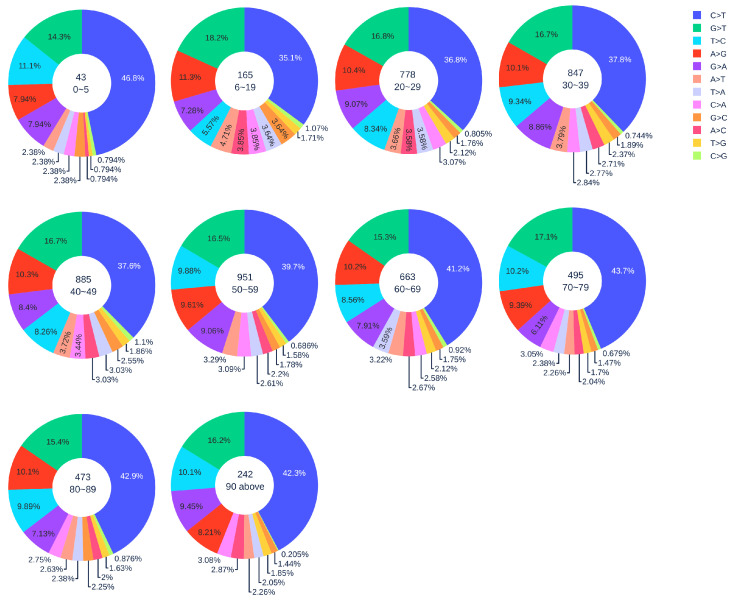
The distribution of 12 SNP types among unique mutations in the SARS-CoV-2 genome isolates from different age groups among female patients. The text inside each circle represents the total number of records that have age information for different age groups.

**Figure 4 viruses-12-01095-f004:**
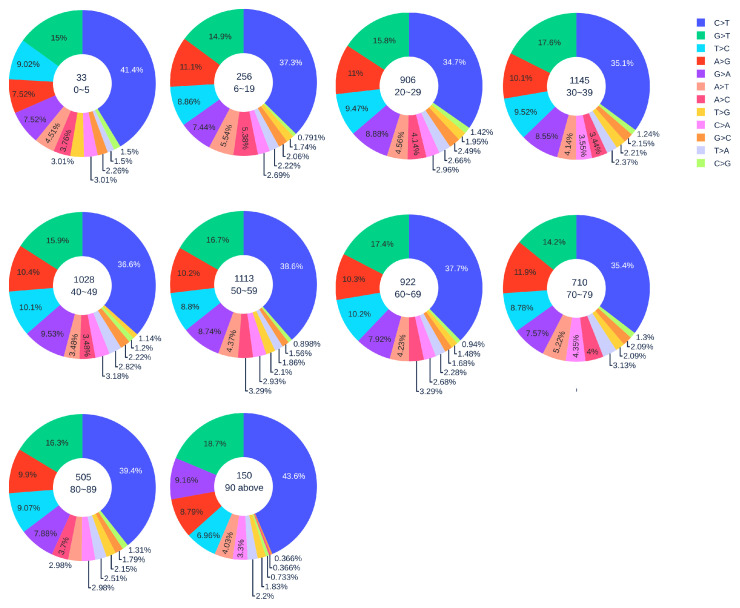
The distribution of 12 SNP types among unique mutations in the SARS-CoV-2 genome isolates from different age groups among male patients. The text inside each circle represents the total number of records that have age information for different age groups.

**Figure 5 viruses-12-01095-f005:**
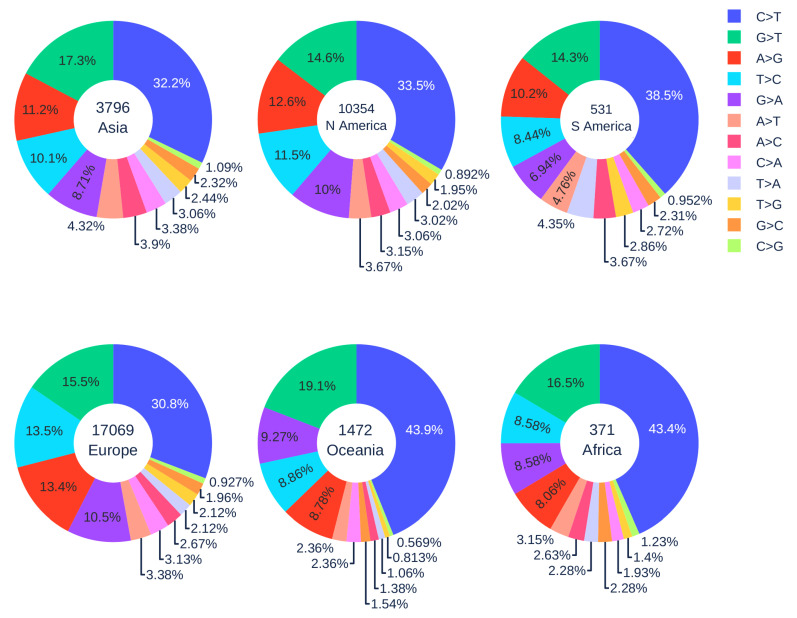
The distribution of 12 SNP types among unique mutations in the SARS-CoV-2 genome isolates in six continents. The text inside each circle represents the total number of records in each continent.

**Figure 6 viruses-12-01095-f006:**
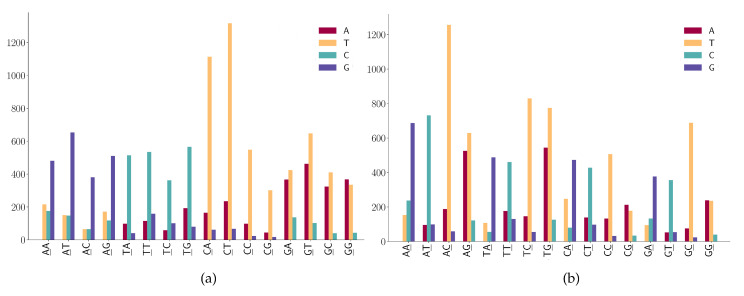
SNP frequencies on 2-mer motifs. (**a**) SNP frequencies are at the first position of 2-mer motifs. (**b**) SNP frequencies are at the second position of 2-mer motifs.

**Figure 7 viruses-12-01095-f007:**
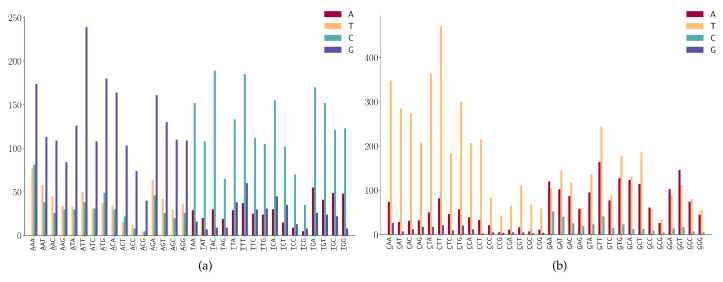
SNP frequency at the first positions of the 3-mer motifs. (**a**) A or T is at the first position of 3-mer motifs. (**b**) C or G is at the first position of 3-mer motifs.

**Figure 8 viruses-12-01095-f008:**
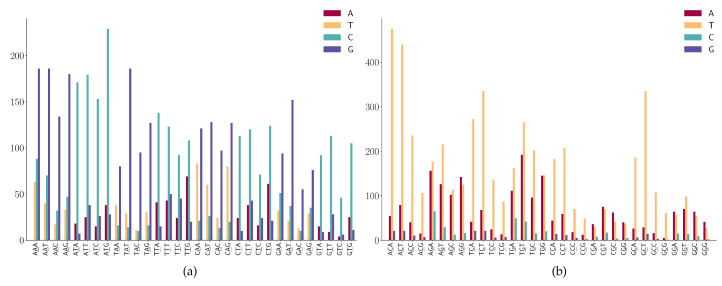
SNP frequency at the second position of 3-mer motifs. (**a**) A or T is at the second position of 3-mer motifs. (**b**) C or G is at the second position of 3-mer motifs.

**Figure 9 viruses-12-01095-f009:**
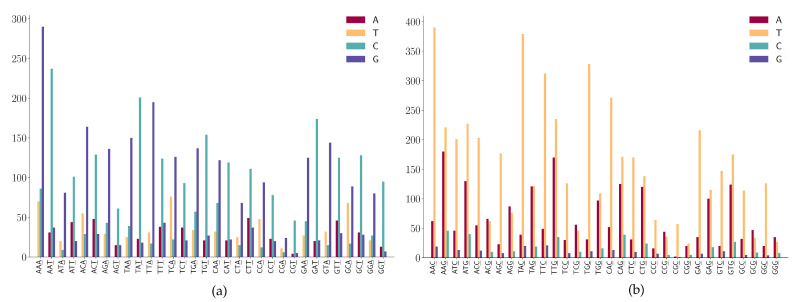
SNP frequency at the third position of the 3-mer motifs. (**a**) A or T is at the third position of 3-mer motifs. (**b**) C or G is at the third position of 3-mer motifs.

**Figure 10 viruses-12-01095-f010:**
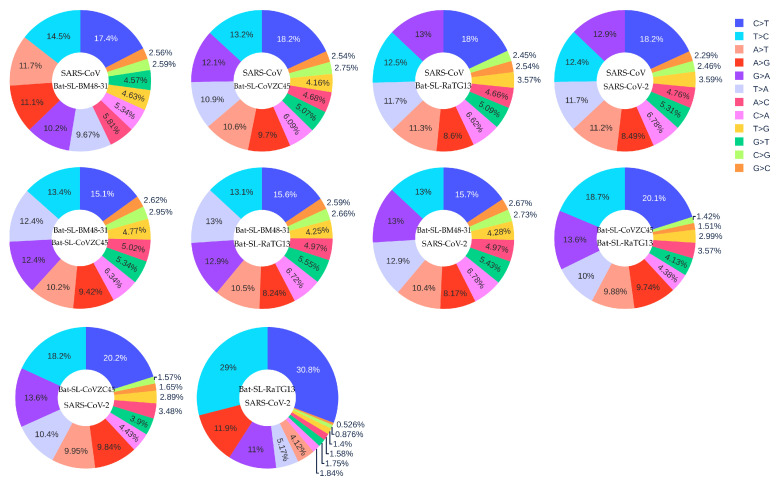
The distribution of 12 SNP types among SARS-CoV, Bat-SL-BM48-31, Bat-SL-CoVZC45, Bat-SL-RaTG13, and SARS-CoV-2. Here, the text at the top represents the reference genome and the text at the bottom represents the mutant sequence.

**Figure 11 viruses-12-01095-f011:**
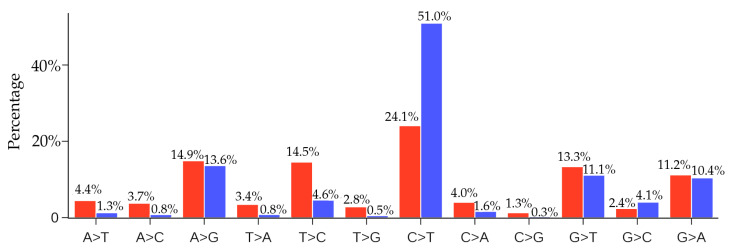
Comparison of the ratios of 12 SNP types among unique mutations (red) and non-unique mutations (blue) in SARS-CoV-2 genomes globally. Here, if we count the same mutation that appears in different SARS-CoV-2 isolates only once, we call those mutations unique mutations. If we count the same mutation in different SARS-CoV-2 isolates repeatedly according to their frequency, then all of the mutations that are detected in the complete SARS-CoV-2 genome sequences are called non-unique mutations.

**Figure 12 viruses-12-01095-f012:**
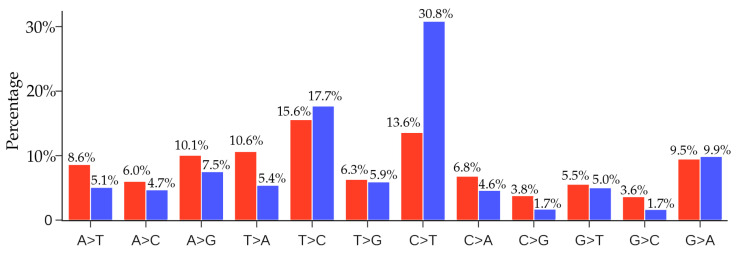
Comparison of the ratios of 12 SNP types among unique mutations (red) and non-unique mutations (blue) in MERS-CoV genome isolates. Here, if we count the same mutation that appears in different MERS-CoV isolates only once, we call those mutations unique mutations. If we count the same mutation in different MERS-CoV isolates repeatedly according to their frequency, then all of the mutations that are detected in the complete MERS-CoV genome sequences are called the non-unique mutations.

**Figure 13 viruses-12-01095-f013:**
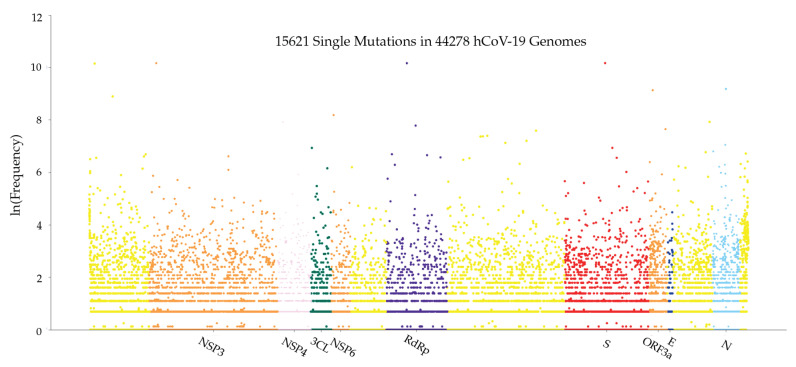
Genome-wide SARS-CoV-2 mutation distribution. The *y*-axis represents the natural log frequency for each mutation on a specific position of the complete SARS-CoV-2 genome. While only a few landmark positions are labeled with gene (protein) names, the relative positions of other genes (proteins) can be inferred from Table 5.

**Table 1 viruses-12-01095-t001:** The distribution of 12 SNP types among unique mutations in the SARS-CoV-2 genome isolates worldwide. The unique SNP mutations are considered in the calculation, i.e., the same type of mutation in different genome isolates is only counted once.

SNP Type	Mutation Type	Ratio	SNP Type	Mutation Type	Ratio
A > T	Transversion	4.44%	C > T	Transition	24.06%
A > C	Transversion	3.75%	C > A	Transversion	4.00%
A > G	Transition	14.87%	C > G	Transversion	1.25%
T > A	Transversion	3.43%	G > T	Transversion	13.33%
T > C	Transition	14.53%	G > C	Transversion	2.36%
T > G	Transversion	2.80%	G > A	Transition	11.17%

**Table 2 viruses-12-01095-t002:** The number of complete SARS-CoV-2 genomes with age/gender information in the United Kingdom, United States, Australia, India, and the world.

Country	Total Counts	Age Counts	Gender Counts
United Kingdom	10,740	2159	2134
United States	8729	1888	2095
Australia	1329	776	750
India	1088	1068	1071
World	33,693	12,513	12,181

**Table 3 viruses-12-01095-t003:** The C > T ratios in different age groups in the United Kingdom, United States, Australia, India, and the world.

Country	0–19	20–29	30–39	40–49	50–59	60–69	70–79	80 above
United Kingdom	44.4%	48.4%	48.7%	49.8%	46.9%	46.6%	48.0%	41.7%
United States	51.0%	45.4%	44.1%	40.0%	41.6%	40.7%	47.6%	45.6%
Australia	35.8%	43.1%	44.1%	41.6%	45.5%	41.7%	45.6%	42.0%
India	39.0%	38.2%	38.2%	35.2%	42.4%	40.9%	46.7%	55.0%

**Table 4 viruses-12-01095-t004:** The C > T over T > C ratios among SARS-CoV (2003), Bat-SL-BM48-31 (2008), Bat-SL-CoVZC45 (2017), Bat-SL-RaTG13 (2013), and SARS-CoV-2 (2019).

From	To	C > T Ratio	T > C Ratio	C > T/T > C Ratio
SARS-CoV-2 reference genome	33693 SARS-CoV-2 genomes	24.06%	14.53%	1.66
SARS-CoV	Bat-SL-BM48-31	17.40%	14.50%	1.20
SARS-CoV	Bat-SL-CoVZC45	18.20%	13.20%	1.37
SARS-CoV	Bat-SL-RaTG13	18.00%	12.50%	1.50
SARS-CoV	SARS-CoV-2	18.20%	12.40%	1.47
Bat-SL-BM48-31	Bat-SL-CoVZC45	15.10%	13.40%	1.13
Bat-SL-BM48-31	Bat-SL-RaTG13	15.60%	13.10%	1.19
Bat-SL-BM48-31	SARS-CoV-2	15.70%	13.00%	1.21
Bat-SL-CoVZC45	Bat-SL-RaTG13	20.10%	18.70%	1.07
Bat-SL-CoVZC45	SARS-CoV-2	20.20%	18.20%	1.11
Bat-SL-RaTG13	SARS-CoV-2	30.80%	29.00%	1.06

**Table 5 viruses-12-01095-t005:** C > T ratios on 26 SARS-CoV-2 proteins. For each gene, the “corrected C > T ratio” is computed by dividing its C > T ratio with the ratio of C in the gene.

Gene Type	Gene Site	Gene Length	C > T Ratio	Corrected C > T Ratio
NSP1	266:805	540	29.0%	1.35
NSP2	806:2719	1914	26.1%	1.41
NSP3	2720:8554	5835	25.4%	1.52
NSP4	8555:10054	1500	30.7%	1.73
NSP5(3CL)	10055:10972	918	32.7%	1.84
NSP6	10973:11842	870	23.7%	1.43
NSP7	11843:12091	249	29.0%	1.47
NSP8	12092:12685	594	28.4%	1.58
NSP9	12686:13024	339	34.4%	1.82
NSP10	13025:13441	417	30.4%	1.51
NSP11	13442:13480	39	33.3%	1.86
RNA-dependent-polymerase	13442:16236	2796	25.4%	1.42
Helicase	16237:18039	1803	26.5%	1.42
3’-to-5’ exonuclease	18040:19620	1581	27.1%	1.48
endoRNAse	19621:20658	1038	19.8%	1.37
2’-O-ribose methyltransferase	20659:21552	894	24.3%	1.52
Spike protein	21563:25384	3819	21.3%	1.13
ORF3a protein	25393:26220	825	22.8%	1.08
Envelope protein	26245:26472	225	22.4%	1.14
Membrane glycoprotein	26523:27191	666	26.4%	1.21
ORF6 protein	27202:27387	183	16.6%	1.19
ORF7a protein	27394:27759	363	21.9%	1.04
ORF7b protein	27756:27887	129	8.3%	0.67
ORF8 protein	27894:28259	363	18.1%	1.03
Nucleocapsid protein	28274:29533	1257	24.0%	0.96
ORF10 protein	29558:29674	114	28.4%	1.58

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
