# Peer review of "Host Immune Response Driving SARS-CoV-2 Evolution"

_viruses, 2020, doi:10.3390/v12101095_

Round 1

Reviewer 1 Report

General

The authors report on the mutation patterns of the human SARS-CoV-2 coronavirus known to be related to Covid-19 disease. Given its world-wide spread the research is timely and relevant to human health. Large (>30,000) number of genomes analyzed seem to be a major strength of the study. It is reported the mutation rate of SARS-CoV-2 is biased towards C>U transitions and that these transitions are more frequent in elderly than in young individuals. While the C>U mutation bias of SARS-CoV-2 has already been reported (those works should be cited) the correlation with age and geographical occurrence are novel and significant findings. In addition, the authors hypothesize that cellular nucleotide deaminases represent the significant source of virus mutability and that their activities might be influenced by the age of an individual. The data are robust, the presentation and conclusions with few exceptions (see below) adequate. Below, please find several points that should be addressed. 

Specific comments

  1. The finding of C>U biases underlying the SARS-CoV-2 evolution has also been reported in two solid works (Matyasek and Kovarik: Genes 11(7):761, 2020. doi: 10.3390/genes11070761.; Simmonds: Sphere 5, e00408-20, 2020, doi: 10.1128/mSphere.00408-20. I feel that the results should be discussed in context of findings in these works and the works cited.
  2. Figures 1,3 and 4. There is something wrong with counting of SNPs in the group of young (0-5 yrs) children. The C>U rates were high in a group of <5 year-old females (46.8%) and males (41.4%) (Figs. 3,4). However, in Figure 1 the total counts (males + females) was 39.6% while simple math says that the value should fall between 41 and 46%. Please, explain.
  3. Figure 15 – The y-axis legend is missing. The values should be given in percentages and not the proportions to fit the data. Further, I wonder how the values in this Figure correspond to those in other figures? For example, the rate of unique C>T mutations in Fig. 15 is 24.1% in Figure 15 while it is about 33% (average of male and female counts) in Fig 2.
  4. Line 190. “Moreover, the reversed mutation type T>C for samples from children under five years old and adults older than 90 years old has the second-largest ratio”. This is not entirely accurate. As follows from Fig 1 the T>C are the third most frequent substitutions (after G>T) in children and elderly. In other groups, the T>C are the fourth most frequent substitutions.
  5. Line 184. I am not convinced whether the “the immune response is stronger in older people“. It is known that the weakening immune system in elderly is the cause of many fatal diseases including cancer.I presume that authors meant that the immune system is reacting rather inadequately in elderly. Please, revise the sentence.
  6. Line 155. The sentence “Moreover, the second most frequent mutation type is A>G transition“. The frequency of A>G is almost the same as that of T>C transitions (Table 1). I doubt that the difference is significant.
  7. Line 159. The sentence “…SARS-CoV-2 fights back the destructive gene editing using its defective proofreading and repairing mechanisms.“ Why should the gene editing be destructive? For example, the C>U mutations have apparently helped to shape the ACE-2 receptor binding site of SARS-CoV-2 (Genes 11(7):761, 2020) suggesting that adaptation of a virus to a new host could be mediated by host RNA editing processes. As with all genetic changes mutations induced by RNA editing should be viewed in context of Darwinian evolution. Most mutations are neutral (not affecting protein composition), some are destructive and only few are beneficial. Hence I see no difference between spontaneous mutations and those arising from RNA editing processes. It is the natural selection which is decisive as to whether a mutation would become beneficial or not. Together, I am not convinced that the gene editing is always destructive.
  8. Line 213. Figure 2 legend.“The text inside each circle represents for the total number of records“ I see the percentages only and not total number of records in the pie charts.
  9. Line 349 The sentence “four transition types (i.e., A>C, C>A, C>T, and T>C)“. I feel that A>C, C>A are transversions and not transitions. Shouldn’t it read A>G and G>A?
  10. The number of Figures in the main text is too high complicating reading. Maybe that some of them could be moved to Supplements, e.g. Figures 5-8, 12-13 and 15).
  11. Methods are only briefly described. In my opinion bioinformatic procedures leading to SNP identification could be better described. Further, only the complete genomes of high-coverage that have no stretches of ’NNNNN’ include in the dataset. How the sequences were filtered for “no Ns“?
  12. Table 1 could be constructed more precisely with horizontal lines and a better design. Alternatively it can be incorporated into Figure 1.

Minor issues

Line 448: SARA>SARS

Line 117: Toreveal>To reveal

It should be mentioned that the accession NC_045512.2 used as a reference is derived from the human coronavirus Wuhan-1 isolate.

Line 418: standing at 13.6> standing at 13.6%?

Author Response

Thank you very much for your email of 1-Sep-2020, and referees reports concerning our manuscript entitled: “Host immune response driving SARS-CoV-2 evolution (Manuscript ID.: viruses-919335). These comments are all valuable and very helpful for revising and improving our paper, as well as important guiding significance to our researchers. We have carefully gone through referees comments, and have revised the manuscript accordingly. The responses are as following:

Reviewer 1:

Comments: The authors report on the mutation patterns of the human SARS-CoV-2 coronavirus known to be related to Covid-19 disease. Given its world-wide spread the research is timely and relevant to human health. Large (>30,000) number of genomes analyzed seem to be a major strength of the study. It is reported the mutation rate of SARS-CoV-2 is biased towards C>U transitions and that these transitions are more frequent in elderly than in young individuals. While the C>U mutation bias of SARS-CoV-2 has already been reported (those works should be cited) the correlation with age and geographical occurrence are novel and significant findings. In addition, the authors hypothesize that cellular nucleotide deaminases represent the significant source of virus mutability and that their activities might be influenced by the age of an individual. The data are robust, the presentation and conclusions with few exceptions (see below) adequate.Below, please find several points that should be addressed.Answer: We thank the reviewer for positive/supportive comments.1. The finding of C>U biases underlying the SARS-CoV-2 evolution has also been reported in two solid works (Matyasek and Kovarik: Genes 11(7):761, 2020. doi: 10.3390/genes11070761.; Simmonds:Sphere 5, e00408-20, 2020, doi: 10.1128/mSphere.00408-20. I feel that the results should be discussed in context of findings in these works and the works cited.Answer: Thank you! We have introduced these two works (marked in red color) in the introduction and cited them.2. Figures 1,3 and 4. There is something wrong with counting of SNPs in the group of young (0-5 yrs)children. The C>U rates were high in a group of<5 year-old females (46.8%) and males (41.4%) (Figs.3,4). However, in Figure 1 the total counts (males + females) was 39.6% while simple math says that the value should fall between 41 and 46%. Please, explain.Answer: There are two reasons.1. The total number of sequences with only age information is 80. However, the total number of sequences with both age and gender information is 76 (33 male and 43 female). Therefore, the C>U rates in the group of<5 year-old will not fall between the rate in a group of<5 year-old males (41.4%) and in a group of<5 year-old females (46.8%).2. Even if the total number of sequences with only age information equals the total number of sequences with age and gender information. The total number of SNPs in the group with only age information will not equal the summation of the total number of SNPs in the male and female groups because we consider the unique SNPs. For example:Sample 1 Age 4, Female, 241C>T, 23403A>G, 25563G>T Sample 2 Age 2, Female, 23403A>G Sample 3 Age 3, Male, 23403A>G, 25563G>T1

Then the total number of unique SNPs for Sample 1-3 is 3, total number of unique SNPs in female group is 3, total number of unique SNPs in Male group is 2.3. Figure 15 The y-axis legend is missing. The values should be given in percentages and not the proportions to fit the data. Further, I wonder how the values in this Figure correspond to those in other figures? For example, the rate of unique C>T mutations in Fig. 15 is 24.1% in Figure 15 while it is about 33% (average of male and female counts) in Fig 2.Answer: They-axis legend has been added. The values are given in percentages. Since we only take the unique SNPs into consideration, the average of male and female rates is not equal to the rates in original Fig. 15 (We have put some figures in to the Supporting Information so the current figure number is different.).4. Line 190. Moreover, the reversed mutation type T>C for samples from children under five years old and adults older than 90 years old has the second-largest ratio. This is not entirely accurate. As follows from Fig 1 the T>C are the third most frequent substitutions (after G>T) in children and elderly. In other groups, the T>C are the fourth most frequent substitutions.Answer: Thanks for pointing this out. We have changed ”second” to ”third”.5. Line 184. I am not convinced whether the the immune response is stronger in older people. It is known that the weakening immune system in elderly is the cause of many fatal diseases including cancer. I presume that authors meant that the immune system is reacting rather inadequately in elderly. Please,revise the sentence.Answer: Thanks! We have revised this sentence and highlight it in red color.6. Line 155. The sentence Moreover, the second most frequent mutation type is A>G transition. The frequency of A>G is almost the same as that of T>C transitions (Table 1). I doubt that the difference is significant.Answer: We have analyzed 33693 complete SARS-CoV-2 sequences and found 13833 unique single mutations.If we focus on the unique mutations, the ratios of A>G and T>C mutation are almost the same. But if we consider the non-unique mutations, the ratios of A>G is 13.6%, while the ratio of T>C is 4.6% (See Fig 15.)Therefore, the difference between A>G mutation and T>C is significant in the population level.7. Line 159. The sentence SARS-CoV-2 fights back the destructive gene editing using its defective proof-reading and repairing mechanisms. Why should the gene editing be destructive? For example, the C>U mutations have apparently helped to shape the ACE-2 receptor binding site of SARS-CoV-2(Genes 11(7):761, 2020) suggesting that adaptation of a virus to a new host could be mediated by host RNA editing processes. As with all genetic changes mutations induced by RNA editing should be viewed in context of Darwinian evolution. Most mutations are neutral (not affecting protein composition), some are destructive and only few are beneficial. Hence I see no difference between spontaneous mutations and those arising from RNA editing processes. It is the natural selection which is decisive as to whether a mutation would become beneficial or not. Together, I am not convinced that the gene editing is always destructive.Answer: Thanks for pointing this out. We did not mean that gene editing is always destructive. We have removed the word ”destructive” in case of any further misunderstanding. The C>U mutation will have a high possibility to terminate the translation of viral proteins since all three stop codons. TAA, TAG, and TGA involve T but not C. Moreover, as mentioned in Matyasek and Kovarik: Genes 11(7):761, 2020. doi:10.3390/genes11070761, C>T mutations in SARS-CoV-2 variants can increase the frequency of codons for hydrophobic amino acids, which may have a dramatic effect on the properties of SARS-CoV-2 proteins. It is in this sense that gene editing appears to be mostly destructive. We have removed the word destructive in our revised version.2

Reviewer 2 Report

viruses-919335
Host immune response driving SARS-CoV-2 evolution
Journal: Viruses

In this paper the authors explore the how mutations could be driving the response strength of SARS-Cov2 in different age groups, gender and countries. The paper is well written but needs to be re-structured in a way that the message comes across more easily. A major drawback of the paper is lack of statistical support for their results.

  • Line 60: “If we consider the average number of mutations per genome, SARS-CoV-2 is mutating slower than other viruses, such as the flu and common cold viruses.” I found this statement really useful to understand the motivation. However, I believe you still need to reference it at this point. Also, I would like to further understand how the T>C C>T, etc mutations rate compare to other viruses.
  • Paragraph from lines 76 – 93: I found this paragraph really convoluted, I read it at least 5 times to understand the message, it needs to be rewritten.
  • Line 101: Missing a space between in and the number.
  • Lines 118 -125: You investigated 12 SNP types in 33693 genome sequences and found that C>T mutations are more common than other mutations. How does it compare to other viruses? At this point I would like to see any statistical evidence to support your finding.
  • Lines 125/126: Please add a reference that support your statement.
  • It is interesting that the position of the SNP in the genome was not considered, I would expect that those mutations are also related to the position in the SARS-CoV2 genome, and not just an overall pattern…
  • Figures are not informative at all. Please, do not use pie charts / donut plots for more than 6 categories. If you want to show the similarities or differences of the different trends here, you could try a line plot or even a barplot. 
  • I’m not sure if considering a uniform distribution for the SNP type ratio makes sense. In many species it is known that some mutations have higher chances of happening than others.
  • Table 1 needs to have a better formatting. Also, you can help the reader by pointing out the transition and transversion mutation types.
  • For all analysis related to Age, Country and Gender it is important to show the statistical significance for your finding before you make the discussions.
  • I would have started the paper with session 2.3, that motivates these differences from other species, and them show the implication of those mutations in humans.
  • Summarize the outcomes from lines 364 – 384 in a Table. It is more elegant and easier to understand than the text.
  • Figure 15: Please fix the numbers on the top. Examples, A>G 14.9% has a box around it, T>G as well. A>G 13.6% the box is overlapping w the bar, and “eats” part of the bar. (You could plot it using ggplot2, for example, where the % would be on top, centralized and not on top of your bars).
  • I would like to see an analysis of the position of those SNPs and how it implicated in the age/gender/country outcomes, this could be easily accessed using GWAS, and would enrich significantly the results. 

Author Response

Thank you very much for your email of 1-Sep-2020, and referees reports concerning our manuscript entitled: “Host immune response driving SARS-CoV-2 evolution (Manuscript ID.: viruses-919335). These comments are all valuable and very helpful for revising and improving our paper, as well as important guiding significance to our researchers. We have carefully gone through referees comments, and have revised the manuscript accordingly. The responses are as following:

Reviewer 2:

In this paper the authors explore the how mutations could be driving the response strength of SARS-Cov2 in different age groups, gender and countries. The paper is well written but needs to be re-structured in a way that the message comes across more easily. A major drawback of the paper is lack of statistical support for their results.Answer: We thank the reviewer for the comments.1. Line 60: If we consider the average number of mutations per genome, SARS-CoV-2 is mutating slower than other viruses, such as the flu and common cold viruses. I found this statement really useful to understand the motivation. However, I believe you still need to reference it at this point. Also, I would like to further understand how the T>C C>T, etc mutations rate compare to other viruses.Answer: Citations have been added in our manuscript. The mutation rates for SARS-CoV-2, MERS, HCoV-OC43, and HCoV229E are listed. (Bar-On, Yinon M., et al. ”Science Forum: SARS-CoV-2 (COVID-19)by the numbers.” Elife 9 (2020): e57309., and Wang, Yanqun, et al. ”Current understanding of middle east respiratory syndrome coronavirus infection in human and animal models.” Journal of thoracic disease 10.Suppl19 (2018): S2260.). We have also added the analysis for 12 types of SNP of MERS-CoV in the discussion section.SARS-CoV-2 mutation rate:103per site per year. For MERS-CoV the mutation rate in the complete genome was estimated to be1.12×103substitutions per site per year, while HCoV-OC43 and HCoV229E represent an average mutation rate of about36×104substitutions per site per year.2. Paragraph from lines 76 93: I found this paragraph really convoluted, I read it at least 5 times to understand the message, it needs to be rewritten.Answer: Thanks! We have modified this paragraph.3. Line 101: Missing a space between in and the number.Answer: Our original file does have a space. We are not sure why the space is missing once we submit to the journal.4. Lines 118 -125: You investigated 12 SNP types in 33693 genome sequences and found that C>T mutations are more common than other mutations. How does it compare to other viruses? At this point I would like to see any statistical evidence to support your finding.Answer: We have added the analysis of MERS-CoV in the discussion. We have also the how these viral mutations should be compared.5. Lines 125/126: Please add a reference that support your statement.Answer: We have added the references.6. It is interesting that the position of the SNP in the genome was not considered, I would expect that those mutations are also related to the position in the SARS-CoV2 genome, and not just an overall1 pattern

Answer: We have added a table (Table 5) in the manuscript, which lists the C>T ratios on 26 SARS-CoV-2proteins. To be noted, we actually consider the position in our manuscript. For example, in the A>G category,we have 187A>G, 2480A>G, 23425A>, 23403A>G.......7. Figures are not informative at all. Please, do not use pie charts / donut plots for more than 6 categories. If you want to show the similarities or differences of the different trends here, you could try a line plot or even a barplot.Answer: Thanks for your suggestion. We have moved some pie charts to the Supporting Information. Addi-tionally, we add some barplots.8. I m not sure if considering a uniform distribution for the SNP type ratio makes sense. In many species it is known that some mutations have higher chances of happening than others.Answer: Yes, some mutations have higher chances of happening than others. As we stated in the Section 2.1.1,the transversion mutation has a relatively low ratio, while the transition mutation has a relatively high ratio.This happens due to the fact that it is easier to substitute a single ring nucleotide structure for another single ring nucleotide structure than to substitute a double ring nucleotide for a single ring nucleotide.9. Table 1 needs to have a better formatting. Also, you can help the reader by pointing out the transition and transversion mutation types.Answer: Thanks for your suggestion. We have changed the Table 1.10. For all analysis related to Age, Country and Gender it is important to show the statistical significance for your finding before you make the discussions.Answer: All of our statements are based on the statistical analysis. We only consider the countries with more than 1000 valid sequences that carry age and gender information. Therefore, we think our analysis is statistical significance. We would appreciate if the reviewer give us a detailed guidance on which specific statistical analysis we should employ further.11. I would have started the paper with session 2.3, that motivates these differences from other species,and them show the implication of those mutations in humans.Answer: We thank the reviewer for the suggestion. Section 2.3 uses some preliminary results/observation from Section 2.1 (i.e, the distribution of 12 SNP types in 33693 complete genome isolates). Therefore, we believe it would be better to put Section 2.3 after Section 2.1.12. Summarize the outcomes from lines 364 384 in a Table. It is more elegant and easier to understand than the text.Answer: We thank the reviewer for the useful suggestion. We have summarize the outcomes from line 354-384in a Table.13. Figure 15: Please fix the numbers on the top. Examples, A>G 14.9% has a box around it, T>G as well.A>G 13.6% the box is overlapping w the bar, and eats part of the bar. (You could plot it using ggplot2,for example, where the % would be on top, centralized and not on top of your bars).

Answer: Thanks! We have fixed it.14. I would like to see an analysis of the position of those SNPs and how it implicated in the age/gender/country outcomes, this could be easily accessed using GWAS, and would enrich significantly the results.Answer: We have created a website that record the position of SNPs SARS-CoV-2 Mutation Tracker (https:// users.math.msu.edu/ users/ weig/ SARS-CoV-2MutationTracker.html), and the screenshot of our mutation2 tracker is placed in the manuscript as well (See Figure 13). Moreover, we have added a table (Table 5) in the manuscript, which lists the C>T ratios on 26 SARS-CoV-2 proteins. Our paper focuses on the over-all statistics analysis of the gene-editing-driven mutations. Analysis on how the position implicated in the age/gender/country outcomes does not appear to reveal further information related to gene-editing.3

Round 2

Reviewer 2 Report

The paper has improved, most of my suggestions have either been incorporated or justified. However, I still cannot find statistical support for any comparison.

Major:

Lack of statistical support. I cannot find any reference to any hypothesis testing of any kind in the paper, for example, when stating that (line 132) “Its ratio of 14.87% A>G is 133 much higher than the average ratio of 8.3%” you need statistical support, it can be that there is no statistical difference, it can be that you find statistical support for it. For that, a simple proportion test would work.

Once you state that there is a difference in the mutation rates from random (that would be a uniform distribution), and you state that there an underling distribution for the mutations, you compare all the next results towards it, and not to the 8.3%.

Lines 177 – 178 “The ratio of C>T mutations in females is slightly higher in males, which matches the finding that women have a stronger immune response than men”, you also do not provide any statistical support. I find it hard to trust the results based only on looking into the raw proportions.

This should be applied to the complete paper, not only in those two examples.

There is an error on the supplementary file, I am unable to download it, therefore I cannot access the new plots. However, the suggestion of changing the visualization of those graphs did not appear in the manuscript, plenty of data viz studies show that our eyes do not perceive more than 6 categories at a time using pie charts (or donuts) plots, therefore, using another visual would benefit highly readers. 

Minor

Line 132: transbition
